⊘ | **Open Peer Review** | Human Microbiome | Methods and Protocols
# Metagenomic estimation of absolute bacterial biomass in the mammalian gut through host-derived read normalization

Gechlang Tang,[1,2] Alex V. Carr,[1] Crystal Perez,[1,3,4] Katherine Ramos Sarmiento,[1] Lisa Levy,[5] Johanna W. Lampe,[5] Christian Diener,[6] Sean M. Gibbons[1,3,7,8,9]

**ABSTRACT** Absolute bacterial biomass estimation in the human gut is crucial for understanding microbiome dynamics and host-microbe interactions. Current methods for quantifying bacterial biomass in stool, such as flow cytometry, quantitative polymerase chain reaction (qPCR), or spike-ins, can be labor-intensive, costly, and confounded by factors like water content, DNA extraction efficiency, PCR inhibitors, and other technical challenges that add bias and noise. We propose a simple, cost-effective approach that circumvents some of these technical challenges: directly estimating bacterial biomass from metagenomes using bacterial-to-host (B:H) read count ratios. We compared B:H ratios to the standard methods outlined above, demonstrating that B:H ratios are useful proxies for bacterial biomass in stool and possibly in other host-associated substrates. B:H ratios in stool were correlated with bacterial-to-diet (B:D) read count ratios, but B:D ratios exhibited a substantial number of outlier points. Host read depletion methods reduced the total number of human reads in a given sample, but B:H ratios were strongly correlated before and after host read depletion, indicating that host read depletion did not reduce the utility of B:H ratios. B:H ratios showed expected variation between health and disease states and were generally stable in healthy individuals over time. Finally, we showed how B:H and B:D ratios can be used to track antibiotic treatment response and recovery. B:H ratios offer a convenient alternative to other absolute biomass quantification methods, without the need for additional measurements, experimental design considerations, or machine learning, enabling robust absolute biomass estimates directly from stool metagenomic data.

**IMPORTANCE** In this study, we asked whether normalization by host reads alone was sufficient to estimate absolute bacterial biomass directly from stool metagenomic data, without the need for synthetic spike-ins, additional experimental biomass measurements, or training data. The approach assumes that the contribution of host DNA to stool is more constant or stable than biologically relevant fluctuations in bacterial biomass. We find that host read normalization is an effective method for detecting variation in gut bacterial biomass. Absolute bacterial biomass is a key metric that often gets left out of gut microbiome studies, and empowering researchers to include this measure more broadly in their metagenomic analyses should serve to improve our understanding of host-microbiota interactions.

**KEYWORDS** metagenomics, human microbiome, gut microbiome, absolute biomass, host DNA, diet DNA

The mammalian gut is a diverse and dynamic ecosystem, containing microorganisms from all domains of life, including archaea, bacteria, viruses, and eukaryotes (1). Bacteria are the most abundant microbes in the gut, by mass, reaching densities of $10^{11}$–$10^{12}$ cells per gram of stool and making up between 25 and 54% of stool dry weight

**Editor** Ákos T. Kovács, Universiteit Leiden, Leiden, Netherlands

**Peer Reviewers** Christoph Kaleta, Christian-Albrechts-Universität zu Kiel, Kiel, Germany; Quinten R Ducarmon, Leids Universitair Medisch Centrum, Leiden, Netherlands

Address correspondence to Sean M. Gibbons, sgibbons@isbscience.org.

S.M.G. is a paid member of the scientific advisory board for Thorne. Thorne was not involved in the work presented in this paper.

See the funding table on p. 17.

(2, 3). The gut microbiota confers essential biomolecular functions to the host (4, 5). Disruption to this ecosystem, as in the case of antibiotic treatment (6), can increase susceptibility to opportunistic infections and other diseases (2, 4). We know that the composition of the gut microbiota is shaped by a combination of intrinsic and extrinsic host factors, such as host genotype, physiology, immunity, behavior, and diet (7–9). Diet and behavior appear to exert the strongest influence (8). Gut microbiome composition is commonly quantified using shotgun metagenomic sequencing of fecal DNA, which provides relative, but not absolute, abundance estimates for microbial taxa and genes (10). Prior work has suggested that accurate estimates of absolute microbial biomass in the gut are crucial to fully understanding cross-sectional and longitudinal variation in this important ecosystem (11–14).

Metagenomic shotgun sequencing is a cost-effective approach to comprehensively quantifying the ecological composition and functional potential of the gut microbiome (2, 15). Standard methods for quantifying absolute biomass require additional measurements beyond the metagenome (16, 17). For example, flow cytometry of dilute stool homogenates can be used to estimate the number of cells per gram of feces (18). Cytometry can be labor-intensive, requiring a dedicated cytometer and extensive standardization, in part due to the large amount of non-cellular debris and caustic compounds present in stool. Additionally, qPCR can be leveraged to detect the total copy number of the 16S gene per gram of stool (or some other marker gene), but PCR can be noisy and sensitive to inhibitors that are common in stool homogenates (19). Furthermore, qPCR absolute abundances and machine-learning approaches that integrate total DNA concentrations from fecal extracts are influenced by fecal DNA extraction efficiency (e.g., samples with lower extraction efficiency will appear to have lower biomass) (17, 20, 21). Spike-ins of DNA (post-extraction) or cells (pre-extraction) from organisms that are not normally present in the system can be used to renormalize relative gut bacterial abundances and obtain absolute biomass estimates (16). Spike-ins are excellent solutions to absolute biomass quantification, but they require additional sample processing steps and result in a reduced number of reads derived from the sample. Finally, recent machine-learning approaches to predict microbial load in stool metagenomes directly from bacterial taxonomic profiles have relied on cytometric biomass estimates as the gold standard, require training data, and achieve somewhat marginal correlation coefficients with out-of-sample microbial load estimates ($R = 0.5$–$0.6$) (22). Most of the biomass estimates listed above are calculated per unit wet weight (i.e., weight of a fresh sample, including the weight of the water in that sample), as opposed to dry weight (i.e., weight is taken before and after drying the sample in an oven, so water weight can be subtracted), so that these microbial biomass estimates are generally conflated with fecal water content (17, 20). However, total fecal biomass in the gut may vary independently of fecal water content. In summary, standard methods for estimating absolute biomass require additional measurements, experimental design considerations, and extensive training data and can suffer from confounding and bias.

Alternatively, some have argued for the use of standard reference frames applied directly to compositional data (10). Specifically, methods have emerged that use log-ratios of different microbiome features to break the underlying compositionality of the data and circumvent the need to estimate total microbial biomass (10, 23). These methods are not dissimilar to spike-ins (e.g., dividing one value by another and taking the log), but leverage features that are already measured in the context of the metagenomic data. The downside to many of these log-ratio methods is that the resulting features become more difficult to interpret. In the simplest case of an additive log ratio, a single taxon is used to normalize the relative abundances of other taxa in the sample, but it is unclear which common denominator taxon should be used as a "control." Here, we propose using the relative abundance of host DNA as that common denominator in stool metagenomic data, dividing the number of bacterial read counts by the number of host read counts to generate a bacteria-to-host (B:H) read ratio. The key assumption is that the average rate of host DNA shedding into stool is relatively constant within

and across healthy individuals, and the extent to which it varies is highly correlated with commensal microbial biomass, allowing us to treat host DNA as a naturally occurring, systemic spike-in. However, the degree to which this assumption holds true across a range of conditions will require scrutiny. Here, we compare ln(B:H) ratios to paired biomass measures derived from flow cytometry, qPCR, and synthetic spike-ins from a number of published studies (4, 16, 24–26). We assess whether or not B:H ratios are associated with fecal water content or stool consistency. We look at the variation in B:H ratios within and across healthy individuals and individuals with a range of diseases. We assess whether B:H ratios are related to bacterial-to-diet (B:D) read ratios. Importantly, we investigate whether or not host read depletion (i.e., a procedure commonly applied to fecal metagenomes prior to uploading them to public repositories) impacts the utility of B:H ratios. Finally, we assess how well stool B:H ratios capture known bacterial biomass trajectories following antibiotic treatment in both mice and humans. We conclude that normalization by host read fraction provides a useful estimate of absolute bacterial biomass in stool metagenomes from healthy individuals, without the additional expense or effort of flow cytometry, qPCR, synthetic spike-ins, or machine learning.

## RESULTS

### Confounding between bacterial load and stool water content

Estimates of gut bacterial biomass are often made per unit wet weight (e.g., cells or 16S copies per gram of fresh stool), unlike bacterial biomass estimates in many other environmental systems, such as soils, which are often normalized to grams dry weight (27, 28). Fresh stool samples can vary substantially in water content, depending on intestinal transit time, with constipation associated with reduced water content and diarrhea associated with elevated water content (29). The Bristol stool scale provides an ordinal score representing stool consistency, with lower scores (1-2) representing hard stools and higher scores (6-7) representing loose stools (30). Bacterial cell density per unit wet weight has been observed to be inversely proportional to stool water content (31), which suggests that current standard estimates of gut bacterial biomass are conflated with water content and intestinal transit time (22). It may be that longer residence times in the colon lead to higher microbial biomass levels (e.g., due to having more time to grow on dietary and host substrates), independent of water content, but standard metrics are not independent of water content and water content is rarely measured. As a thought experiment, let us imagine a scenario where the total bacterial biomass of a given bolus of stool remains fixed, while water content and total stool mass can vary along the Bristol scale (Fig. 1a). If a standard amount of fresh stool (e.g., 200 mg) is sampled for biomass quantification (e.g., cytometry or DNA extraction), there will be an inverse relationship between cell count or 16S copy number and water content (Fig. 1b). However, if cell counts were normalized to dry weight, there should be no differences along the Bristol scale (Fig. 1c). Below, we explore the interdependence between fecal water content, traditional measures of biomass that are normalized by the wet weight of stool, and B:H ratios.

### The impact of host read depletion on B:H read count ratios

Many human microbiome studies run bioinformatic host read depletion on fecal metagenomes (32, 33) to reduce the amount of host genomic information in raw FASTQ files prior to uploading them to a public repository. This poses a potential challenge to using B:H ratios as biomass estimates in host read-depleted samples. To investigate the impact of host read depletion on B:H ratios, we leveraged a data set that we generated in-house from the Gut Puzzle cohort ($n = 39$). We ran the raw FASTQ files and the host read-depleted FASTQs through our standard metagenomic processing pipeline (see Materials and Methods). As expected, we saw a significant drop in the human read fraction in the host read-depleted samples (two-sided Welch's $t$-test, $P < 0.001$; Fig. 2a). We observed strong correlations between paired human read fractions (Fig. 2b) and B:H

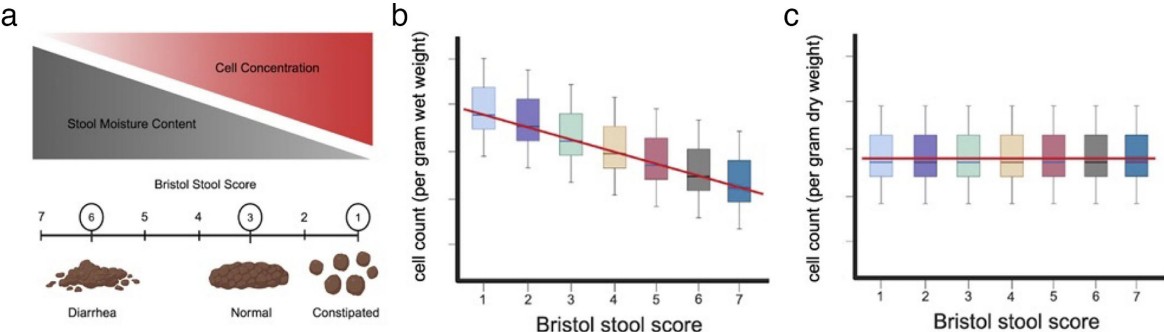

**FIG 1** Conceptual figure explaining the potential confounding between stool moisture content and biomass estimates. (**a**) A schematic of how moisture content and cell concentration per unit wet weight are related. Here, stool consistency is measured by the Bristol stool score (i.e., a proxy for water content), ranging from 1 (constipated; low moisture) to 7 (diarrhea; high moisture). Constipated stools (1–2) exhibit a high density of bacterial cells per gram but contain low moisture, while loose stools (6–7) have higher moisture levels with lower bacterial cell density per gram wet weight. (**b**) Due to the inverse relationship between cell concentration and stool water content, cell count estimates per unit wet weight of stool will be negatively associated with the Bristol score, even if total biomass does not vary. (**c**) If we assume that total biomass per bolus of stool does not vary across a set of samples that span the Bristol scale, cell counts per unit dry weight will show no differences along the Bristol scale.

ratios (Fig. 2c) before and after host read depletion (Pearson's $r = 0.94$, $P < 0.001$, for both). These results indicate that the quantitative nature of host read fraction variation across samples is maintained in host read-depleted samples.

## Comparing B:H ratios, microbial load measures, stool consistency measures, and B:D read count ratios

In the MetaCardis cohort ($N = 850$), we found a subtle, but significant, positive association was observed between ln(B:H) ratios and log cytometric bacterial biomass estimates (i.e., microbial load), after adjusting for covariates that showed significant associations with either B:H ratios or microbial load values (Table S1), which included age, gender, nationality, body mass index, mapped read count, fasting plasma triglycerides, waist circumference, fasting plasma adiponectin, and left ventricular ejection fraction (adjusted linear regression, $r^2 = 0.022$, $P = 0.027$; Fig. 3a). Covariates that showed significant associations with B:H ratios were limited to three cardiometabolic health markers: fasting triglycerides, fasting adiponectin, and waist-to-hip ratios. Microbial load showed significant associations with mapped read count, nationality, fasting triglycerides, and left ventricular ejection fraction. While the fasting triglycerides covariate was common to both microbial biomass metrics, only microbial load showed significant (adjusted linear regression, $P < 0.05$) associations with potential proxies for technical or batch effects (i.e., nationality and mapped read count).

In order to look at the relationship between bacterial biomass estimates and stool moisture content, we pulled down existing data from Vandeputte et al. (18), which included paired measures of cytometric bacterial biomass (microbial load) and moisture content from 223 stool samples (18). We observed a significant inverse association between microbial load (cells per gram of fresh stool) and percent stool moisture content (linear regression, $r^2 = 0.14$, $P < 0.001$; Fig. 2b). We were unable to calculate B:H ratios for the same set of samples because Vandeputte et al. generated 16S amplicon sequencing data, rather than metagenomes. In order to obtain paired measures of stool consistency (Bristol stool scores; a proxy for stool water content) and B:H ratios, we used available data from the 39 Gut Puzzle participants (Fig. 2c). We saw no significant association between the log B:H ratios and Bristol scores in this cohort (ordinal logistic regression, $P = 0.441$). In this same cohort, we saw no significant association between the human read fraction and Bristol scores (ordinal logistic regression, $P = 0.418$; Fig. S1).

Another source of potential non-bacterial reads that could be used for biomass normalization in the gut is those reads coming from the host diet. Our group recently

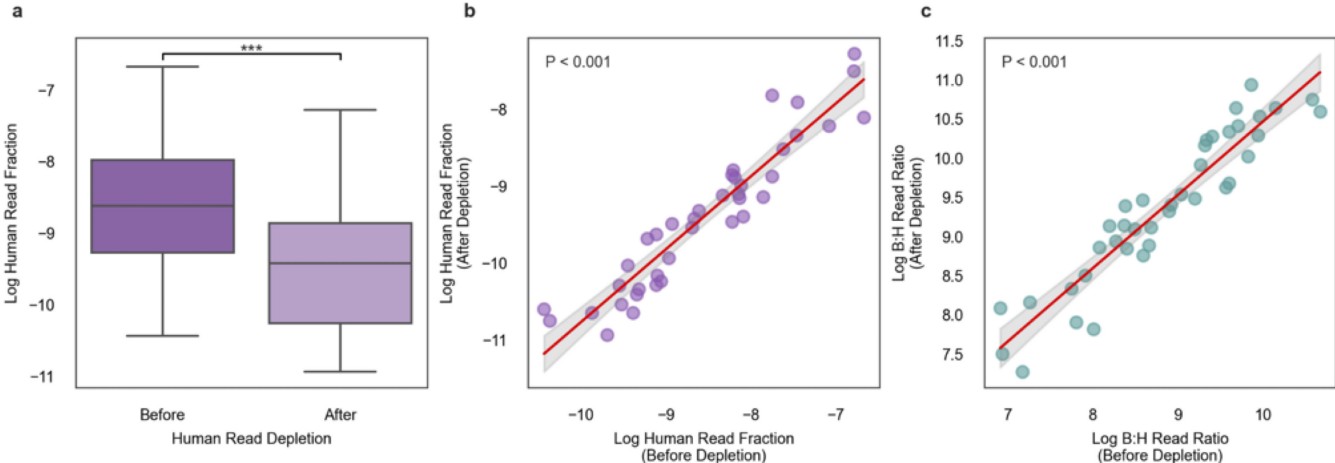

**FIG 2** The impact of human read depletion on B:H ratios in the Gut Puzzle cohort. (a) Box plots comparing log human read fractions (relative to total metagenomic reads) before and after depletion, showing a statistically significant difference using a two-sided *t*-test (***$P < 0.001$; $n = 39$). Each boxplot displays the center line (median), box limits (first and third quartiles), and whiskers (1.5 × interquartile range). (b) Pearson correlation showing a strong correlation ($r = 0.94$, $P < 0.001$; $n = 39$) between human read fractions before and after human read depletion. (c) Pearson correlation showing a strong correlation between log-transformed B:H read ratios before and after human read depletion ($r = 0.94$, $P < 0.001$; $n = 39$).

developed a method for extracting these dietary reads from human fecal metagenomes, called MEDI (34). After applying MEDI to the MetaCardis cohort, we observed a statistically significant correlation between B:H ratios and B:D ratios (Pearson's $r = 0.220$, $P < 0.001$; Fig. 3d), although there was a bimodal distribution of B:D ratios (i.e., several outliers) that reduced the strength of this association. This bi-modality in B:D ratios was driven by a set of samples with very low amounts of food reads detected, which could be related to processed food intake (i.e., we tend to see fewer food reads detected in individuals with higher processed food intake) (34). B:H ratios showed a more normal distribution across samples, indicating that B:H ratios may be preferable for biomass estimation.

### B:H ratios in mice show quantitative agreement with qPCR estimates of total bacterial biomass and with dietary read-based biomass normalization

In data obtained from Chng et al. (4), we found that log-transformed B:H ratios in mice, derived from shotgun metagenomic data, were significantly associated with log-transformed absolute 16S rRNA gene copies quantified by qPCR across 107 fecal pellets ($r^2 = 0.656$, $P < 0.001$; Fig. 4a). We also observed a significant association between log-transformed B:H ratios and total bacterial biomass estimates normalized by log-transformed B:D read ratios from metagenomic sequencing data, across 242 mouse fecal pellets from the same study, in which the mice were maintained on a standard chow diet ($r^2 = 0.718$, $P < 0.001$; Fig. 4b). In summary, we see strong agreement between B:H ratios, qPCR-based bacterial biomass estimates, and B:D ratios (i.e., normalized to plant-derived reads, which are likely from the diet; this diet normalization was reported as an alternative absolute biomass estimation approach in the original Chng et al. paper) in mouse stool.

### Comparing synthetic and natural spike-ins for biomass estimation in milk metagenomes

We had trouble identifying an appropriate stool spike-in data set with a high enough sample number. As an alternative approach, we pulled down data from Wallace et al. (16), encompassing metagenomic data from 385 cow milk samples with controlled bacterial spike-ins (i.e., a specific microbe that was known to be absent from milk). We observed a strong positive association between log-transformed B:H ratios and

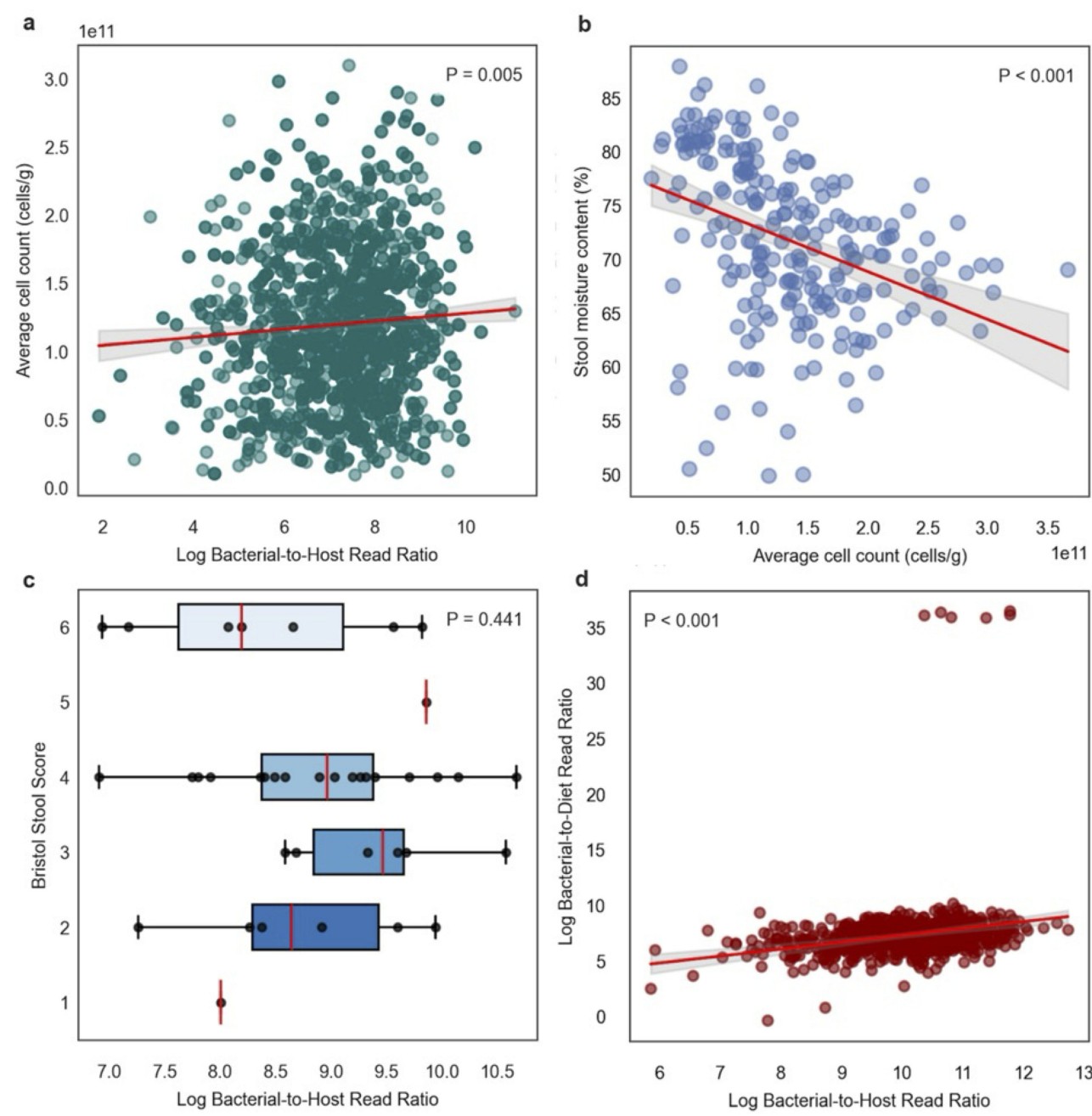

**FIG 3** Comparing cell counts per gram of fresh stool (microbial load), moisture content (Bristol score), B:D read count ratios, and B:H ratios. (**a**) Residual plot showing a statistically significant positive adjusted association between log-transformed B:H ratios and bacterial cell counts per gram of wet stool ($n = 850$), after adjusting for relevant covariates (see Table S1). The red line represents the linear regression fit between residuals ($R^2 = 0.022$, $P = 0.027$) using data derived from Fromentin et al. (26). The y-axis represents bacterial load of stool samples (cells per gram) as estimated by flow cytometry, displayed on a log scale (1e11 = 1 × $10^{11}$). (**b**) Scatterplot showing a statistically significant negative association between bacterial cell counts per gram of wet stool (microbial load) and stool moisture content ($n = 223$). The red line represents the linear regression fit ($R^2 = 0.140$, $P < 0.001$) of data obtained from Vandeputte et al. (18). (**c**) Boxplots showing B:H ratios across Bristol stool score categories ($n = 39$). Each boxplot displays the center line (median), box limits (first and third quartiles), and whiskers (1.5 × interquartile range). Using ordinal logistic regression, we did not observe a significant association between B:H ratios and Bristol scores ($P = 0.441$). (**d**) Pearson's correlation plot showing a statistically significant correlation between log-transformed B:H ratios and log-transformed B:D ratios ($r = 0.220$, $P < 0.001$; $n = 741$) using data from Fromentin et al. (26).

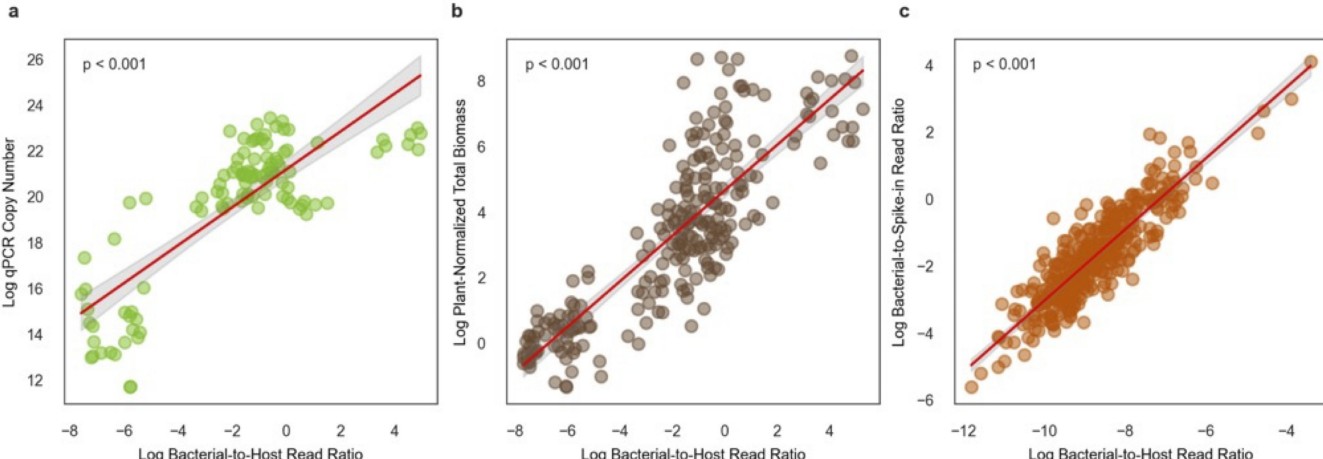

**FIG 4** Associations between B:H read count ratios, 16S qPCR-based bacterial biomass estimates, diet-read normalized bacterial biomass estimates, and spike-in normalized bacterial biomass estimates. (**a**) Scatterplot depicting a significant positive association between log-transformed B:H read ratios and qPCR-quantified biomass in mouse stools (log 16S copy number per microliter; $n = 107$). The red line represents the linear regression fit ($R^2 = 0.656$, $P < 0.001$). (**b**) Scatterplot depicting a significant positive association between log-transformed B:H read ratios and shotgun sequencing-based, diet-normalized total bacterial biomass estimates in mouse fecal samples (normalized by plant-derived reads present in the stool metagenome; $n = 242$). The red line represents the linear regression fit ($R^2 = 0.718$, $P < 0.001$). (**c**) Scatterplot illustrates a significant positive association between log-transformed B:H read ratios and log-transformed endogenous-to-spike-in read ratios from metagenomic sequencing of cow milk samples ($n = 385$). The red line represents the linear regression fit ($R^2 = 0.784$, $P < 0.001$). Data for (**a**) and (**b**) were obtained from Chng et al. (4), and data for (**c**) were obtained from Wallace et al. (16).

log-transformed total endogenous bacteria-to-spike-in ratios ($r^2 = 0.784$, $P < 0.001$; Fig. 3c). This robust association supports the concept that host reads serve as a naturally occurring spike-in that can be leveraged to estimate absolute bacterial biomass in metagenomic data sets from other host-associated substrates beyond stool.

## Examining intra- and inter-individual variation in bacterial and human relative DNA abundances in stool from healthy individuals and individuals with a number of disease conditions

In data obtained from Poyet et al. (35), we were able to observe day-to-day fluctuations in B:H ratios, bacterial relative abundances, and host read relative abundances across four healthy individuals with long, dense stool metagenomic time series (Fig. 5a through f). Inter-individual differences in the mean ln(B:H) were significant between all donors, except for between donors am and ao (two-sided Welch's *t*-test, $P < 0.05$; Fig. 5b). Similar patterns were seen for the relative abundances of human and bacterial reads across these four donors (Fig. 5c through f). The largest fold difference in average B:H ratios across these four healthy donors was 2.4, between donors "an" and "am" (Fig. 5b).

We next investigated how B:H ratios varied across health and disease states in the MetaCardis and integrated human microbiome project 2 (iHMP2 [36]) cohorts (Fig. 6). If we look at the interquartile range (i.e., to exclude outliers) for microbial load and B:H ratios in the MetaCardis cohort, we see ~19-fold and ~3-fold variation, respectively, providing a range of expected variation in these metrics across healthy stool donors. We next compared microbial load estimates across health and cardiometabolic disease groups in the MetaCardis cohort (Fig. 6a). We observed a few subtle differences in microbial load between groups that passed our significance threshold, but there were no significant differences between the healthy group and the other groups (analysis of variance [ANOVA], followed by a Tukey's *post hoc* test for inter-group comparisons, alpha = 0.05). Different pairs of groups showed significant differences when looking at the B:H ratios, with a general trend of lower B:H ratios in the healthy group compared to several cardiometabolic disease groups (ANOVA, followed by a Tukey's *post hoc* test for inter-group comparisons, alpha = 0.05; Fig. 6b). While we did not necessarily expect strong perturbations to total gut bacterial biomass across cardiometabolic diseases, we did

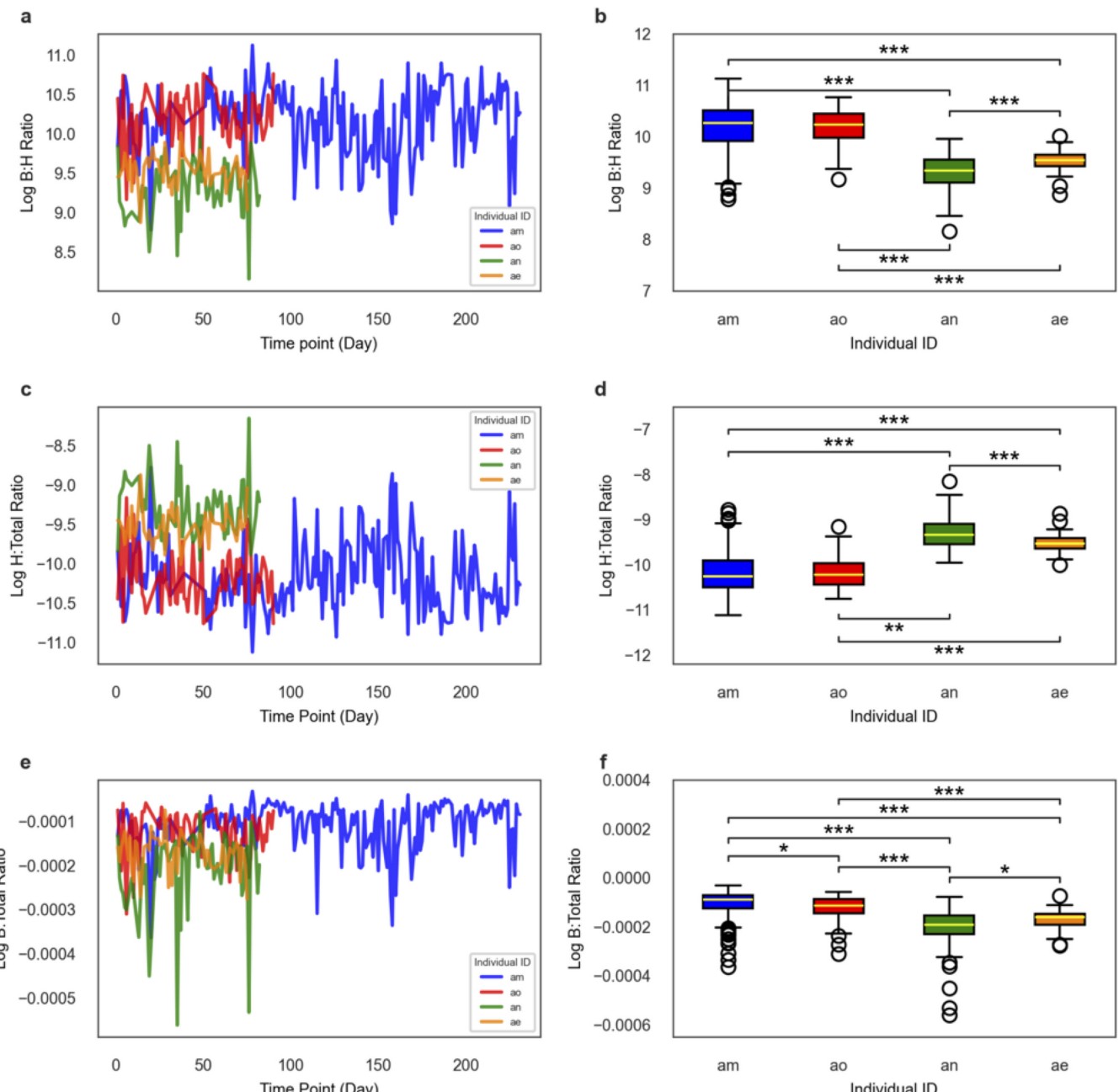

**FIG 5** Temporal dynamics of bacteria-to-host ratios in healthy humans. (**a**) Dense time-series plot displaying an overall trend of day-to-day intra-individual fluctuations in log-transformed B:H read ratios, derived from metagenomic sequencing of human stool samples collected from four healthy individuals ($n =$ 205 for donor am; $n =$ 57 for donor ae; $n =$ 62 for donor an; and $n =$ 74 for donor ao). (**b**) Boxplot showing the distributions of log-transformed B:H read ratios across the same four healthy individuals. Each boxplot depicts the median (center line), interquartile range (box limits, representing the first and third quartiles), and whiskers (extending to 1.5 × the interquartile range). (**c, d**) Similar plots to panels a and b, but for human-to-total (H:total) read ratios (normalized by total metagenomic reads from a sample). (**e, f**) Similar plots to panels a and b, but for bacterial-to-total (B:total) ratios (raw $P$-values for the boxplots multiplied by six, which was the number of pairwise comparisons made, prior to applying the alpha <0.05 threshold). For all panels: Bonferroni-corrected ***$P <$ 0.0001, **$P <$ 0.001, *$P <$ 0.05 (two-sided Welch's $t$-test). The data were obtained from Poyet et al. (35).

expect to find a negative relationship between inflammatory bowel disease (IBD) and gut bacterial biomass due to the inflammation and diarrhea that often accompanies this disease (37). Indeed, we saw that ulcerative colitis (UC) and Crohn's disease (CD) cohorts showed significantly lower B:H ratios than healthy controls (HCs), and between the IBD cohorts, UC showed lower B:H ratios than CD (ANOVA, followed by a Tukey's *post hoc* test

for inter-group comparisons, alpha = 0.05; Fig. 6c). We observed a similar pattern when looking at the host read fraction, with healthy controls showing the lowest host read fraction, followed by CD and then UC (Fig. 6d).

## Tracking responses to antibiotic treatment with B:H ratios

As a positive control for our B:H ratio approach to bacterial biomass estimation, we pulled down metagenomic data from two studies by Palleja et al. (6) and Chng et al. (4) that treated humans and mice with antibiotics, respectively, sampling before, during, and after treatment (4, 6). We first applied our metric by plotting the log-transformed B:H ratios from human stool metagenomic data sampled from 12 individuals across five time points: day 0 (baseline), day 4 (during antibiotic intervention), day 8, day 42, and day 180 (post-antibiotic recovery; Fig. 7a). We observed a significant decline in B:H ratios from day 0 to day 4 (two-sided Welch's t-test, $P < 0.001$), indicating significant bacterial biomass depletion due to antibiotics, followed by a rapid recovery to baseline levels by day 8, which persisted throughout the time series (Fig. 7a). Using pairwise t-tests across five time points (days 0, 4, 8, 42, and 180), accounting for false discovery rate, we found a significant increase in host reads during antibiotic treatment (day 4; paired t-test, $P < 0.001$). To further validate our B:H metric, we also looked at B:D ratios across the time series and found significant depletion relative to baseline on day 4, followed by a rapid recovery by day 8 (two-sided Welch's t-test, $P < 0.001$; Fig. 7b) (38).

We observed a similar pattern for log-transformed B:H ratios sampled from 27 mice across nine time points, with a steep drop in bacterial biomass during antibiotic treatment (days 3, 6, and 7; two-sided Welch's t-test, $P < 0.001$), with ratios returning to baseline levels by day 10 (Fig. 7c). These analyses demonstrated consistent patterns of rapid microbiome depletion during antibiotic exposure, with an average ~45-fold drop in B:H ratios in the human cohort and an average ~400-fold drop in the B:H ratios in the mouse data, followed by recovery within several days. Overall, we found that cross-sectional variation in B:H ratios in healthy human samples was less than ~9-fold, while antibiotic-induced drops in B:H ratios in humans are on the order of ~45-fold.

## DISCUSSION

In this study, we evaluated whether normalization by host reads alone was sufficient to estimate absolute bacterial biomass directly from stool metagenomic data, without the need for training machine-learning models, synthetic spike-ins, or additional experimental measurements, like flow cytometry or qPCR. We compared and contrasted B:H ratios to other more established biomass estimation methods and showed how B:H ratios are robust to host read depletion, and we validated B:H ratios using longitudinal data from humans and mice treated with antibiotics.

The Gut Puzzle data set provided an opportunity to calculate B:H ratios for samples with and without host read depletion. We saw the anticipated drop in detected host reads in the read-depleted samples, but we also saw a strong correlation between paired B:H ratios before and after host read depletion (Fig. 2). Given this observation, it appears that B:H ratios can be applied to samples with and without host read depletion. However, we recommend that B:H ratio comparisons be restricted to sets of samples that have gone through the same wet-lab and bioinformatic processing pipelines to avoid batch effects.

We found that stool moisture content was inversely associated with cytometric cell counts per gram of fresh stool (i.e., often termed "microbial load") (31). B:H ratios showed a positive association with microbial load and no clear association with Bristol stool scores (a proxy for water content), indicating that B:H ratios may be more direct measures of bacterial biomass (i.e., independent of stool moisture content; Fig. 3). Prior studies have identified stool moisture content and bowel movement frequency, respectively, as major confounding factors in microbiome analyses (18, 22, 39). As we outline above, stool consistency is not necessarily related to total bacterial biomass in the gut (e.g., a vegan with loose stool could have much higher total bacterial biomass in their gut than a

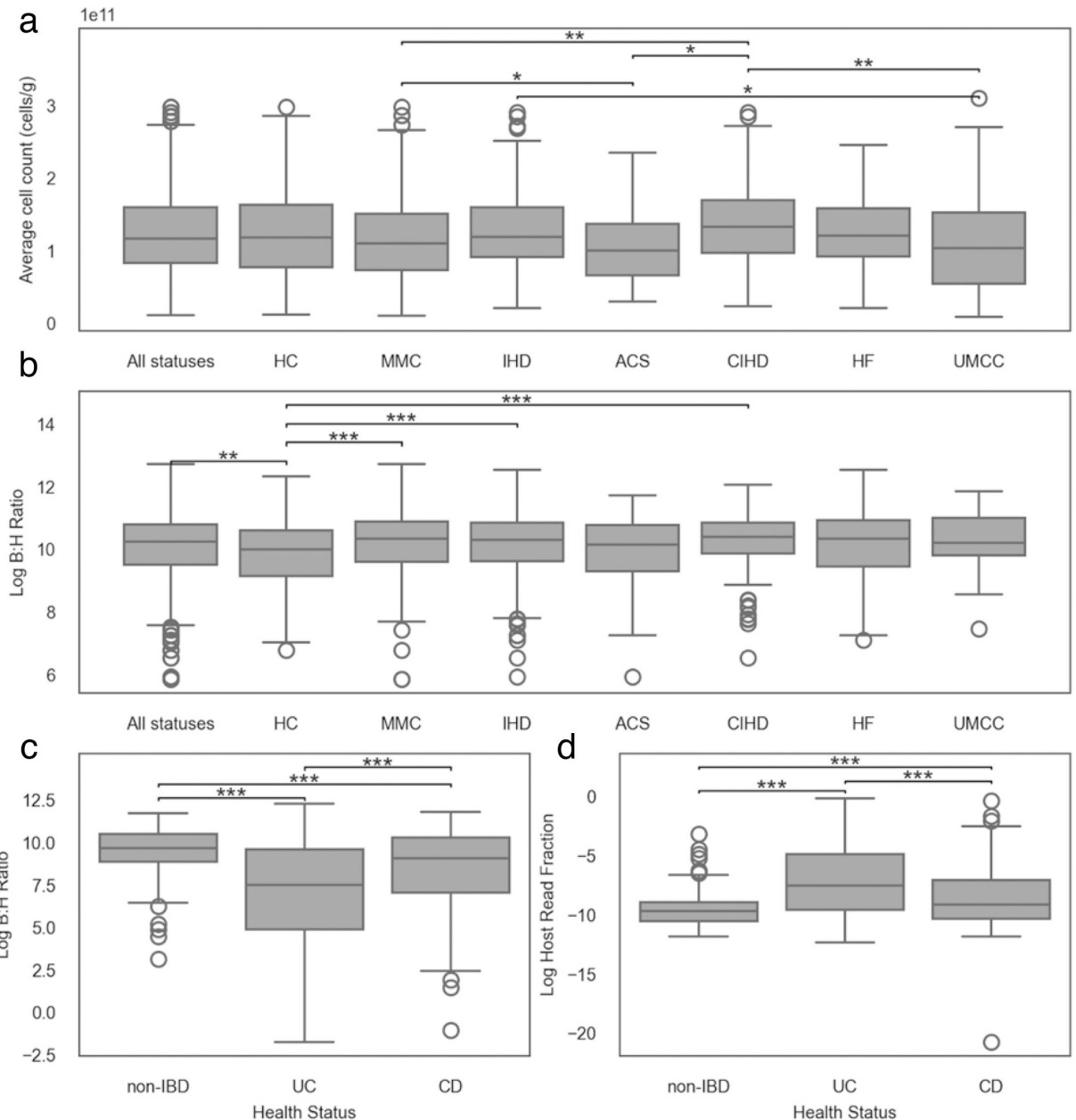

**FIG 6** Box plots of log-transformed microbial load, log-transformed B:H read ratios, and log-transformed human read fractions across health and disease groups in the MetaCardis and iHMP2 cohorts. (**a**) Bacterial cell counts per gram of wet stool across health subgroups in the MetaCardis cohort ($n = 1,305$). (**b**) Log-transformed B:H ratios across health subgroups in the MetaCardis cohort ($n = 1305$): HC, metabolically matched controls (MMC), ischemic heart disease (IHD), acute coronary syndrome (ACS), chronic ischemic heart disease (CIHD), heart failure (HF), and untreated metabolically matched controls (UMMC). (**c**) Log-transformed B:H ratios across health subgroups (non-IBD, UC, CD) in the iHMP2 cohort ($n = 670$). (**d**) Log-transformed host read fractions (relative to total metagenic reads) across health subgroups in the iHMP2 cohort ($n = 670$). Each boxplot displays the center line (median), box limits (first and third quartiles), and whiskers ($1.5 \times$ interquartile range). Statistical significance was determined using ANOVA followed by Tukey's *post hoc* tests (***$P < 0.0001$, **$P < 0.001$, *$P < 0.05$). Data for panels (**a**) and (**b**) were derived from Fromentin et al. (26), and data for panels (**c**) and (**d**) were derived from Lloyd-Price et al. (36).

carnivore with constipation, but this might not be apparent when looking at wet weight-normalized microbial load) (31, 40, 41), and it is important to have biomass estimates that are independent of moisture content and transit time.

We saw strong agreement between qPCR-based estimates of absolute 16S copy number and B:H read ratios in mouse stool (Fig. 3a). In the original study, the authors found that plant-derived reads in the metagenomic data, likely coming from the mouse

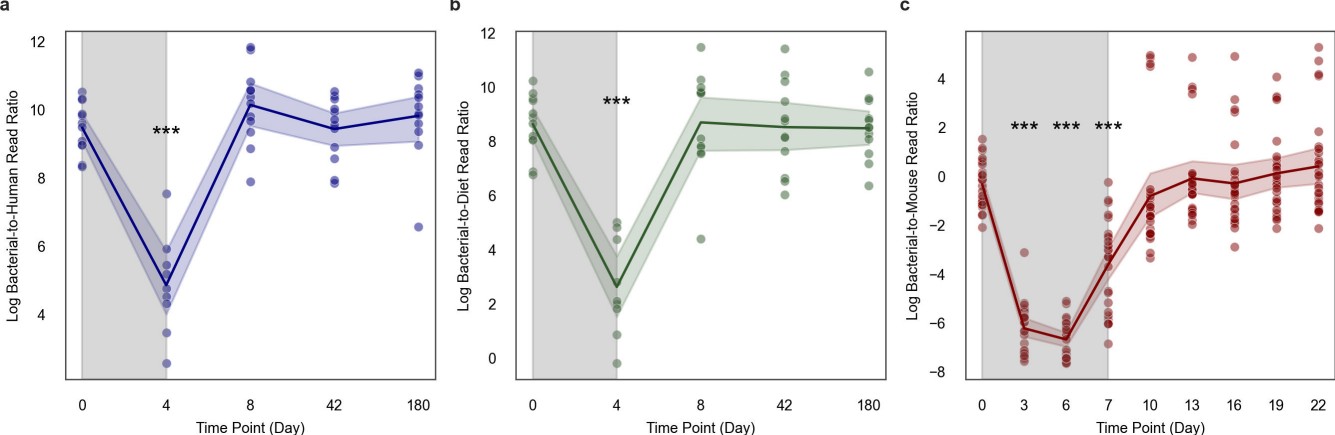

**FIG 7** Temporal dynamics of bacteria-to-host ratios before, during, and after antibiotic treatment in humans and in mice. (**a**) Line plot showing the log-transformed B:H read ratios from metagenomic sequencing of human stools sampled from 12 individuals ($n = 58$) across five time points (days): baseline (day 0), during (day 4), and after (days 8, 42, and 180) antibiotic treatment. An initial sharp decline was observed, followed by a rapid recovery post-antibiotics. The data were obtained from Palleja et al. (6). (**b**) Line plot showing the log-transformed B:D read ratios from the same human cohort as in (**a**). A similar pattern of sharp decline followed by recovery post-antibiotics was observed. (**c**) Line plot showing the log-transformed B:H read ratios from metagenomic sequencing of mouse stools sampled from 27 mice ($n = 243$) over nine time points (days): baseline (day 0), during (days 3-6), and after (days 7, 10, 13, 16, 19, and 22) antibiotic exposure, showing a similar pattern of depletion and recovery. These data were derived from Chng et al. (4). Shading around lines represents 95% confidence intervals, and the gray shaded regions indicate the antibiotic treatment windows.

diet, were inversely related to qPCR biomass estimates. We found that normalizing by both plant-derived and host-derived reads provided roughly equivalent estimates of bacterial biomass (Fig. 3b). Together, these findings reinforce the utility of B:H ratios, and perhaps B:D ratios, for generating accurate estimates of bacterial biomass in the mammalian gut. Unlike mice, however, humans consume a wide variety of diets, and there is evidence that dietary read frequencies in human stool can fluctuate over several orders of magnitude, depending on the types of foods consumed (38). Higher variability in dietary reads is supported by the bimodal distribution of B:D ratios in the MetaCardis cohort (Fig. 3d). As such, host DNA may be a more reliable normalization factor in human stool.

Perhaps the most widely accepted method for biomass normalization in metagenomic sequencing is the spiking-in of a controlled amount of cells or DNA from an organism that is not present in the system (e.g., a hyperthermophile spiked into a stool sample). We looked at the largest host-associated synthetic spike-in data set we could identify ($N = 385$), which consisted of cow milk samples where ZymoBIOMICS Spike-in Control I (High Microbial Load) was added to assess bacterial biomass levels. We found that B:H ratios (using a cow reference genome to quantify host reads, see Materials and Methods) were tightly associated with the bacterial read counts normalized by spike-in counts (Fig. 3c). Synthetic spike-ins require additional experimental design considerations, and they reduce the number of sequencing reads from target organisms. Leveraging natural spike-ins, like host-derived sequences, appears to be sufficient for absolute biomass estimation. Taken together with the fecal metagenomic results, we suggest that host-associated substrates, like stool, milk, vaginal fluid, or saliva, contain a relatively stable amount of host DNA that can be leveraged for absolute bacterial biomass estimation.

Major declines in gut bacterial biomass have been associated with IBD, antibiotic treatment, chemotherapy, and gastrointestinal cancers, while higher bacterial biomass and diversity have been associated with both health and constipation (22, 42). In line with these observations, we saw a significant drop in B:H ratios in IBD patients relative to healthy controls in the iHMP2 (36) cohort, with UC patients showing lower B:H ratios than CD patients (Fig. 6c). Declines in B:H ratios in IBD could be driven either by higher rates of epithelial damage and host DNA shedding or by drops in anaerobic commensal biomass

due to inflammation-induced oxidative stress, although these mechanisms are highly inter-correlated and likely both contribute to the observed phenomenon. Although it is harder to interpret, B:H ratios also revealed subtle but significant increases in B:H ratios in certain cardiometabolic disease categories, compared to healthy controls, in the MetaCardis cohort (Fig. 6b).

As a final assessment of our approach, we analyzed time-series data from healthy human and mouse cohorts that received broad-spectrum antibiotic treatments (Fig. 5). B:H ratios showed a ~45-fold and a ~400-fold drop following antibiotic treatment in humans and mice, respectively, followed by a recovery back to the baseline levels (Fig. 5a and c). These antibiotic-induced shifts in estimated biomass are much larger than the ~2.4-fold difference observed between average B:H ratios across the Poyet et al. (35) metagenomic time series, the ~9-fold differences observed in the human antibiotic cohort at the baseline time point, and the ~3-fold variation in the interquartile range observed in healthy individuals in the MetaCardis cohort, indicating that cross-sectional variation in healthy individuals is substantially smaller than major, clinically relevant disruptions to gut bacterial biomass.

Recent efforts to estimate absolute bacterial abundance in the gut have introduced machine-learning models that leverage DNA concentration and other sample-level metadata to predict total microbial load (21, 22). While these models achieve moderately strong performance in validation cohorts, some depend on access to sample-level DNA concentration values and other metadata that are rarely included in public metagenomic data sets. These models also rely on cytometric or qPCR measures as biomass standards in the training data sets, which, as we discuss above, can be confounded by water content, DNA extraction efficiency, and PCR inhibitors. In contrast, our B:H read ratio method relies solely on information that can be obtained directly from a metagenome. As such, our approach offers a lightweight, metadata-independent solution to estimating absolute bacterial biomass, which makes it well suited for retrospective analyses, without requiring additional experiments, specialized assays, or training data.

A major limitation of our approach is that even if we have a reliable global estimate of total bacterial biomass, we cannot necessarily use it to infer absolute biomass levels for individual taxa. Unknown detection biases for each taxon obscure these taxon-specific absolute abundance estimates (43, 44). However, this limitation also applies to other approaches that attempt to estimate absolute abundances from cytometry, qPCR, or by other means. It is unclear whether or not these feature-wise corrections (e.g., as in the case of "quantitative microbiome profiling") (18), which leverage total community biomass estimates, are useful or appropriate. Despite this limitation, we believe that estimates of total bacterial biomass in the gut will be an informative covariate to include in metagenomic analyses.

In conclusion, the B:H ratio represents a simple approach for estimating absolute bacterial biomass in stool, and possibly in other host-derived substrates, leveraging host read counts that are often disregarded in metagenomic sequencing studies. The B:H ratio appears to be more independent of stool consistency and water content than other stool bacterial biomass measures that are normalized per unit wet weight. Surprisingly, this approach is robust to host read depletion, making it applicable to a wide variety of public data sets. Absolute bacterial biomass is a key metric, and empowering researchers to include this measure more broadly in their metagenomic analyses should serve to improve our understanding of host-microbiota interactions.

## MATERIALS AND METHODS

### Data sources and processing

#### Cohort with paired 16S sequencing data, stool moisture content, and microbial load metadata

Preprocessed data were taken directly from the Supplemental Information section of Vandeputte et al. (18). This data set included bacterial biomass measurements, determined by a C6 Accuri flow cytometer (BD Biosciences) after mechanical homogenization, as well as stool moisture content, which was measured in duplicate as the percentage of mass loss after freeze-drying 0.2 g of frozen, homogenized fecal material stored at −80°C.

#### Gut Puzzle cohort

Fecal samples from 39 Gut Puzzle participants with Bristol Stool Score metadata were collected and processed using our lab's custom Nextflow pipeline (https://github.com/Gibbons-Lab/pipelines/tree/master/metagenomics). Samples were collected in 1,200 mL two-piece specimen collectors (Medline) in the Public Health Sciences Division of the Fred Hutchinson Cancer Center (Institutional Review Board protocol number 10961) and transferred into a large vinyl anaerobic chamber (Coy; 37°C, 5% hydrogen, 20% carbon dioxide, balanced with nitrogen) at the Institute for Systems Biology within 30 min of sample receipt. Fecal aliquots were sent to Diversigen, Inc., for DNA extraction, library preparation, and shotgun metagenomic sequencing. Briefly, libraries were prepared with the Nextera XT DNA Library Preparation kit (Illumina) and sequenced with a paired-end $2 \times 150$ bp protocol on a NovaSeq 6000 (Illumina) yielding at least 70 M reads per sample. Initial quality control was performed using fastp (45), where reads were trimmed to remove low-quality bases, with a minimum quality threshold set at 20, a minimum read length of 50 bp, and a maximum read length of 150 bp to ensure the retention of high-quality data. Taxonomic relative abundances were estimated using Kraken2 (46) and Bracken (47), with a custom Kraken2 database (kraken2_db_uhgg_v.2.0.1 database) constructed using data from Almeida et al. (48), including the human genome (GRCh38 reference assembly). For this analysis, we used a confidence threshold of 0.3 for genus and species-level identification across multiple taxonomic ranks.

Depletion of host reads was accomplished using the host contamination removal tool (HoCoRT) (49) and the *Homo sapiens* GRCh38 reference assembly. HoCoRT was used to build a GRCh38 Bowtie2 index. Raw metagenomics data (FASTQ files) were then filtered by removing reads that mapped to the GRCh38 index using the HoCoRT implementation of Bowtie2 (50). For the index construction and filtering steps, the HoCoRT Bowtie2 default settings were used. Host-depleted metagenomic data were then processed using the pipeline for estimating taxonomic relative abundance described above.

#### Metagenomic data from the European MetaCardis cohort

Raw metagenomic data (FASTQ files) from Fromentin et al. (26) were downloaded from European Nucleotide Archive (ENA) under the accession numbers PRJEB37249, PRJEB38742, PRJEB41311, and PRJEB46098. The data were reprocessed using the same Nextflow-based pipeline described above. The MetaCardis cohort included 869 HCs and individuals across varying stages of dysmetabolism and ischemic heart disease severity, aged 18–75 years, recruited from Denmark, France, and Germany between 2013 and 2015. Bacterial biomass in stool samples was quantified using a C6 Accuri flow cytometer and expressed as cell counts per gram of fecal material (i.e., microbial load index). Fecal DNA was extracted according to the IHMS guidelines (SOP 07 V2 H), and sequencing was performed in a non-randomized order using Ion Proton technology (Thermo Fisher Scientific).

## Mouse antibiotic treatment data set

Raw metagenomic data (FASTQ files) from Chng et al. (4) were downloaded from Sequence Read Archive (SRA) under the accession number SRP142225. The data were reprocessed using the same Nextflow-based pipeline described above, with a modified Kraken2 database that includes the mouse reference genome, which is the Genome Reference Consortium Mouse Build 39, for classification of host reads (51). Taxonomic abundances were estimated using Bracken (47), applying an abundance cutoff of 10 reads before reassignment. Stools were sampled as a cage unit (two mice per cage) over multiple time points: before antibiotic treatment with 2.5 mg/day ampicillin sodium salt administered via oral gavage (day 0); at midpoint of antibiotic treatment (day 3); at the end-point of antibiotic treatment (day 6); and at 1 day post-gavage (day 7), 4 days post-gavage (day 10), 7 days post-gavage (day 13), 10 days post-gavage (day 16), 13 days post-gavage (day 19), and 16 days post-gavage (day 22). Total bacterial DNA was extracted from fecal samples using the PowerSoil DNA isolation kit (MoBio Laboratories) according to the manufacturer's instructions.

Absolute 16S rRNA genes were quantified with qPCR using a pair of universal 16S primers, which can be found in the Supplemental information from Chua et al. (52). DNA from six treatment groups was amplified on days 0, 3, 10, and 13. Each reaction was prepared in triplicate on a 384-well plate, containing 5 µL PowerUp SYBR Green Master Mix, 0.5 µL of 5 µM primers, and 1 µL of 10× diluted DNA, with a total volume of 10 µL. The ViiA 7 Real-Time PCR System (Thermo Fisher Scientific) was used for qPCR with the following amplification specifications: 1 cycle of 95℃ for 2 min and 40 cycles of 95℃ for 15 s, 60℃ for 15 s, and 72℃ for 1 min. A standard curve, created from serial dilutions of synthesized DNA, was used to convert Ct values to copy numbers, and day 0 copy numbers were used to normalize bacterial abundances across samples. These results were directly sourced from the supplement.

Diet-normalized bacterial biomass was estimated by normalizing all reads classified to bacterial taxa against plant-derived reads, under the assumption that the amount of diet-derived plant DNA would be conserved across mouse fecal samples. These data were sourced directly from the supplement.

## Metagenomic data of bulk milk samples

Processed metagenomic data from cow milk samples, including total bacterial reads, total host reads, and total spike-in reads, were obtained from the supplementary materials section from Wallace et al. (16). Bulk milk samples were collected from 276 commercial dairy cows in New Zealand. For these bulk samples, a 15 mL subsample was sent weekly to the Herd Testing facility for host cell counting. All samples were stored at −20℃ prior to DNA extraction and spiked with 17 µL of a 1:100 diluted spike-in control (ZymoBIOMICS Spike-in Control I, High Microbial Load). Following extraction, short-read shotgun sequencing libraries (150 bp paired-end) were prepared using the Illumina DNA Flex library prep kit, and sequencing was performed on an Illumina NovaSeq system with S1 and S4 flow cells, aiming for 15 million reads per sample. Quality control checks were conducted using the FastQC program, and samples with fewer than 100,000 reads were excluded from further analysis. Reads were classified using Kraken2 against a database comprising microbiome, human (GRCh38), and bovine (ARS-UCD1.2) sequences downloaded from NCBI's RefSeq database (53), allowing identification of bacterial, host (bovine), and spike-in reads.

## Metagenomic data of a cohort of 12 healthy humans given a 4-day antibiotic intervention

Raw metagenomic data (FASTQ files) from Palleja et al. (6) were downloaded from ENA under the accession number ERP022986 and reprocessed using the same Nextflow-based pipeline described above. Stool samples were collected from 12 healthy Caucasian men who were 18 to 40 years of age. In addition to a screening visit, the study design

encompassed five study visits (D0, D4, D8, D42, and D180) and a 4-day broad-spectrum antibiotic intervention consisting of once-daily administration of 500 mg meropenem, 500 mg vancomycin, and 40 mg gentamicin dissolved in apple juice and ingested orally. Microbial DNA was extracted from 200 mg frozen stool and sequenced. An average of 79.4 ± 18.0 million raw metagenomic reads per sample was generated, corresponding to 7.94 ± 1.8 Gb of data. The average sequencing depths for samples collected at time points D0, D4, D8, D42, and D180 were 76.5 ± 11.1, 78.1 ± 13.2, 75.6 ± 19.6, 81.2 ± 11.4, and 85.4 ± 26.6 million reads, respectively, indicating no significant reduction in read depths immediately following the intervention. To ensure data quality, reads were subjected to adapter removal and trimmed based on a quality score threshold of 20 and a minimum read length of 30 base pairs. This process resulted in an average of 6.8 million reads being discarded due to adapter contamination, while 0.94 million reads were removed for not meeting the trimming criteria. Human DNA contamination was eliminated by aligning reads against the human genome (version hg19). Approximately 0.24 million reads per sample, excluded during this step, were used to quantify host read counts for subsequent analysis. After these quality control measures, the final data sets contained high-quality non-human reads of 69.3 ± 8.7 million for D0, 66.5 ± 13.1 million for D4, 68.2 ± 15.8 million for D8, 72.0 ± 12.1 million for D42, and 79.8 ± 22.6 million for D180.

### Longitudinal metagenomic data from four healthy individuals in the Broad Institute-OpenBiome Microbiome Library (BIO-ML) cohort

The raw metagenomic data (FASTQ files), including total bacterial read counts, host read counts, and overall read counts, belonging to Poyet et al. (35), were downloaded from SRA under the accession number PRJNA544527 (35) and reprocessed using the same Nextflow-based pipeline as described above. A total of 1,207 stool samples were collected from 90 participants between July 2014 and May 2016 and sourced from the non-profit stool bank OpenBiome. Donors, aged 19 to 45 years (mean age of 28), had body mass indexes from 17.5 to 29.8 (mean of 23.4) and were screened by OpenBiome to ensure they were healthy and pathogen-free. Samples were deidentified, diluted 1:10 in a solution of 12.5% glycerol and 0.9% NaCl, homogenized, and filtered through a 330 µm filter. DNA was extracted using the MoBio PowerSoil 96 kit (Qiagen cat no. 12955-4) with minor modifications. After thawing on ice, 625 µL to 1 mL of homogenized stool was added to the PowerSoil plate (12955-4-BP) and centrifuged at 4,000 $g$ for 10 min. Following removal of the supernatant, 750 µL of bead solution and 60 µL of C1 solution were added. Samples were bead-beaten at 20 Hz for 10 min, rotated 180 degrees, and beaten for an additional 10 min. They were then centrifuged at 4,500 $g$ for 6 min, and 850 µL of the supernatant was transferred to a clean collection plate. The remaining steps followed the manufacturer's protocol. Metagenomic DNA was quantified using the Quant-iT PicoGreen dsDNA Assay (Life Technologies) and normalized to 50 pg/µL. Illumina sequencing libraries were generated from 100 pg–250 pg of DNA with the Nextera XT DNA Library Preparation kit (Illumina), following the manufacturer's protocol with scaled reaction volumes. Libraries were pooled by combining 200 nL from each of 96 samples. Insert sizes and concentrations of the pooled libraries were verified with an Agilent Bioanalyzer DNA 1000 kit (Agilent Technologies). Sequencing was performed on a HiSeq system (2 × 101 bp), targeting ~10 million paired-end reads. Shotgun metagenomic sequencing data underwent quality trimming to remove low-quality bases and human-aligned reads (hg19), followed by duplicate sequence removal using fastuniq. This process yielded approximately $9.8 \times 10^6$ high-quality reads per sample. The filtered reads were assembled with metaSPAdes, and protein-coding genes were identified using Prodigal. To reduce redundancy, genes were clustered with CD-HIT to generate a nonredundant gene set, which was subsequently annotated with Cluster of Orthologous Genes (COG) terms using rps-blast. Finally, Bowtie2 was employed to align the reads to the COG-annotated gene set, and the relative abundances of COG families were determined based on gene coverage.

### Healthy and Inflammatory Bowel Disease Samples from iHMP2

The raw stool shotgun metagenomic data (FASTQ files) and metadata on health and disease status from Lloyd-Price et al. (36) were downloaded from the iHMP2 data portal (https://ibdmdb.org/results) and reprocessed using the same Nextflow-based pipeline as described above. The iHMP2 cohort was recruited to study multi-omic signatures of inflammatory bowel disease (IBD), including non-IBD controls, ulcerative colitis (UC) patients, and Crohn's Disease (CD) patients (n =670).

## STATISTICAL ANALYSES

Due to the right-skewed nature of the B:H ratios, bacteria-to-diet ratios, bacteria-to-spike-in ratios, and qPCR copy numbers, we applied a natural log transformation so that distributions behaved more normally. Pairwise comparisons of the log-transformed B:H, B:total, and H:total ratios among the four donors in the dense time-series analyses were conducted using Welch's $t$-tests, assuming unequal variance. A Bonferroni correction was applied to adjust for multiple testing, with the significance threshold set to $\alpha = 0.05 / N$, where $N$ is the number of pairwise comparisons. Corrected $P$-values were evaluated against this adjusted threshold. Linear regressions were used to assess the relationships between the log-transformed B:H ratios and other bacterial biomass metrics, with the B:H ratios serving as the independent variable. An ordinal logistic regression model was used to examine the relationship between stool consistency (Bristol score) and the B:H ratios, with consistency as the dependent variable. Additionally, line plots were employed to visualize the distributions of log B:H ratios in humans and compare them with bacterial-to-host read ratios in mice. One-way ANOVA was performed to assess differences in microbial load and B:H ratios across clinical subgroups in the MetaCardis cohort, and in B:H ratios and host read fractions in the iHMP2 cohort, followed by Tukey's *post hoc* tests to evaluate pairwise group differences. Pearson correlation was used to assess the relationships between log-transformed B:H ratios and B:D ratios, as well as between human read fractions before and after host read depletion. All statistical analyses and visualizations were conducted using Python 3.9.19 with the following libraries: pandas (1.5.3), numpy (1.26.4), statsmodels (0.14.0), matplotlib (3.8.4), seaborn (0.13.2), scipy (1.12.0), and scikit-learn (1.2.2). See code availability section for analysis notebooks.

## ACKNOWLEDGMENTS

Research reported in this publication was supported by the National Institute of Diabetes and Digestive and Kidney Diseases (NIDDK) of the National Institutes of Health (NIH) under award number R01DK133468 (to S.M.G.). The fecal sample collection at Fred Hutchinson Cancer Center for the Gut Puzzle study was supported in part by P30 CA015704.

## AUTHOR AFFILIATIONS

[1]Institute for Systems Biology, Seattle, Washington, USA

[2]Master of Science Program in Genetic Epidemiology, University of Washington School of Public Health, Seattle, Washington, USA

[3]Molecular Engineering Graduate Program, University of Washington, Seattle, Washington, USA

[4]Medical Scientist Training Program, University of Washington, Seattle, Washington, USA

[5]Fred Hutchinson Cancer Center, Seattle, Washington, USA

[6]Diagnostic and Research Institute of Hygiene, Microbiology and Environmental Medicine, Medical University of Graz, Graz, Austria

[7]Department of Bioengineering, University of Washington, Seattle, Washington, USA

[8]Department of Genome Sciences, University of Washington, Seattle, Washington, USA

⁹University of Washington, eScience Institute, Seattle, Washington, USA

**AUTHOR ORCIDs**

Gechlang Tang  http://orcid.org/0009-0007-2745-7114
Sean M. Gibbons  http://orcid.org/0000-0002-8724-7916

**FUNDING**

| Funder | Grant(s) | Author(s) |
|---|---|---|
| HHS | NIH | National Institute of Diabetes and Digestive and Kidney Diseases (NIDDK) | R01DK133468 | Sean M. Gibbons |
| HHS | National Institutes of Health (NIH) | P30 CA015704 | Johanna W. Lampe |

**AUTHOR CONTRIBUTIONS**

Gechlang Tang, Conceptualization, Formal analysis, Investigation, Methodology, Validation, Visualization, Writing – original draft, Writing – review and editing | Alex V. Carr, Methodology, Software, Supervision, Writing – review and editing | Crystal Perez, Resources, Validation, Writing – review and editing | Katherine Ramos Sarmiento, Resources, Validation, Writing – review and editing | Lisa Levy, Resources, Validation, Writing – review and editing | Johanna W. Lampe, Resources, Validation, Writing – review and editing | Sean M. Gibbons, Conceptualization, Funding acquisition, Project administration, Resources, Supervision, Writing – original draft, Writing – review and editing.

**DATA AVAILABILITY**

Preprocessed 16S amplicon data from Vandeputte et al. (2017) are available in the supplementary information section of that study (18). Preprocessed metagenomic data from Fromentin et al. (2022) are available in the supplementary information section from that study (26). Preprocessed metagenomic data of milk samples by Wallace et al. (2023) are provided in the supplementary data tables of the supplementary material section from that study (16). Raw metagenomic data from Chng et al. (2020) can be found in the Sequence Read Archive (SRA) under project ID SRP142225. Raw metagenomic data from Palleja et al. (2018) are accessible in the European Nucleotide Archive (ENA) under accession number ERP022986. Raw metagenomic data from Poyet et al. (2019) can be found at NCBI BioProject under accession number PRJNA544527. Raw metagenomic data from the iHMP2 cohort can be found on their data portal (https://ibdmdb.org/results). Gut Puzzle data can be found at NCBI BioProject under accession numbers PRJNA1033794 and PRJNA1284320.

Nextflow pipelines for processing metagenomic shotgun sequencing data, from raw reads to taxonomic abundance matrices, are available https://github.com/Gibbons-Lab/pipelines/ (metagenomics pipeline). Scripts used for analyzing the data and generating the figures in this study can be accessed at https://github.com/Gibbons-Lab/Metagenomic_Biomass_Quantification_2024.

**ADDITIONAL FILES**

The following material is available online.

Supplemental Material

**Supplemental information (mSystems00984-25-S0001.docx).** Fig. S1 and Table S1.

Open Peer Review

**PEER REVIEW HISTORY (review-history.pdf).** An accounting of the reviewer comments and feedback.

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
