## [Reviewer comments · mSystems]

Metagenomic estimation of absolute bacterial biomass in the mammalian gut through host-derived read normalization.

Gechlang Tang, Alex Carr, Crystal Perez, Katherine Ramos Sarmiento, Lisa Levy, Johanna Lampe, Christian Diener, and Sean Gibbons

Corresponding Author(s): Sean Gibbons, Institute for Systems Biology

Review Timeline:

Submission Date:

July 1, 2025

Accepted:

July 8, 2025

Editor: Ákos T. Kovács

Reviewer(s): Disclosure of reviewer identity is with reference to reviewer comments included in decision letter(s). The following individuals involved in review of your submission have agreed to reveal their identity: Christoph Kaleta (Reviewer #1); Quinten R Ducarmon (Reviewer #2)

Transaction Report:

DOI: <https://doi.org/10.1128/msystems.00984-25>

Re: mSystems00984-25 (Metagenomic estimation of absolute bacterial biomass in the mammalian gut through host-derived read normalization.)

Dear Prof. Sean M. Gibbons:

Your manuscript has been accepted, and I am forwarding it to the ASM production staff for publication. Your paper will first be checked to make sure all elements meet the technical requirements. ASM staff will contact you if anything needs to be revised before copyediting and production can begin. Otherwise, you will be notified when your proofs are ready to be viewed.

Sincerely,
Ákos T. Kovács
Editor
mSystems

Reviewer #1 (Comments for the Author):

The authors have thoroughly addressed all of my concerns.

Reviewer #2 (Comments for the Author):

Great job on the rebuttal.